# Strong plasmon-molecule coupling at the nanoscale revealed by first-principles modeling

Tuomas P. Rossi [1], Timur Shegai [1], Paul Erhart [1] & Tomasz J. Antosiewicz [1,2]

Strong light-matter interactions in both the single-emitter and collective strong coupling regimes attract significant attention due to emerging applications in quantum and nonlinear optics as well as opportunities for modifying material-related properties. Exploration of these phenomena is theoretically demanding, as polaritons exist at the intersection between quantum optics, solid state physics, and quantum chemistry. Fortunately, nanoscale polaritons can be realized in small plasmon-molecule systems, enabling treatment with ab initio methods. Here, we show that time-dependent density-functional theory calculations access the physics of nanoscale plasmon-molecule hybrids and predict vacuum Rabi splitting. By considering a system comprising a few-hundred-atom aluminum nanoparticle interacting with benzene molecules, we show that cavity quantum electrodynamics holds down to resonators of a few cubic nanometers in size, yielding a single-molecule coupling strength exceeding 200 meV due to a massive vacuum field of $4.5\,\mathrm{V}\cdot\mathrm{nm}^{-1}$. In a broader perspective, ab initio methods enable parameter-free in-depth studies of polaritonic systems for emerging applications.

[1] Department of Physics, Chalmers University of Technology, 412 96 Gothenburg, Sweden. [2] Faculty of Physics, University of Warsaw, Pasteura 5, 02-093 Warsaw, Poland. Correspondence and requests for materials should be addressed to T.J.A. (email: tomasz.antosiewicz@fuw.edu.pl)

L ight-matter interaction between an optical mode and a quantum emitter is accurately described in terms of the cavity quantum electrodynamics (cQED) formalism[1]. The interaction can be weak or strong, depending on circumstances, leading to drastically different behavior. The regime of strong light-matter coupling, unlike its weak counterpart, leads to formation of hybrid cavity-emitter eigenmodes manifested in coherent energy exchange between the subsystems occurring on timescales that are much faster than the corresponding damping rates. Thus, the emitter and the cavity form a unified light-matter hybrid polariton, whose properties, including spontaneous emission and chemical potential, can be tuned[2–4]. Because of their compositional nature, polaritons are useful for photon–photon interactions, which lead to remarkable nonlinear and quantum optical phenomena, including photon blockade and Bose-Einstein condensation of exciton polaritons[5–7]. On the other hand, strong light-matter coupling may lead to changes in emitter properties, including photochemical rates[8–12] and exciton transport[13,14].

Traditionally cavity-exciton polaritons have been realized in atomic or solid state systems utilizing high quality-factor optical cavities at low temperatures[15]. Polaritonic behavior associated with "dressing" of the emitter by a cavity field is usually captured by traditional quantum optical approaches such as Jaynes–Cummings or Dicke models[2–4,16,17]. However, these quantum optical formalisms treat matter in an extremely simplified manner, that is, as a two-level system, leading to oversimplifications and inconsistencies in the description of the material subpart.

Advanced theoretical techniques developed recently allow for more sophisticated effects including, for example, multiple electronic resonances, accounting for atomic vibrations, and light-matter interactions beyond the point-dipole approximation. Significant progress along these lines has been achieved by several groups using various quantum optical and quantum chemistry methods[9–11,18–21]. However, typically either molecules or electromagnetic fields in these approaches are treated in a simplified manner.

Plasmon-molecule interactions can be modeled computationally using density-functional theory (DFT)[22,23] and/or time-dependent density-functional theory (TDDFT)[24] approaches, thanks to the relatively small number of atoms involved in these interactions. Here, we model the entire plasmon-exciton system by TDDFT, which allows the whole system to be treated on the same footing, enabling one to track effects related to modification of the matter subpart, which are inaccessible by purely quantum optical or classical electromagnetism methods. In fact, analytical and numerical quantum chemistry approaches, including DFT and TDDFT, have been successfully applied to study plasmon-molecule interactions, charge transfer, chemical enhancement and electromagnetic effects in surface-enhanced Raman scattering experiments[25–33]. However, vacuum Rabi splitting and strong plasmon-molecule coupling have not been in the focus. Yet, many recent experimental observations cannot be easily explained with current theories[34–36].

In response to the above, quantum electrodynamics density-functional theory (QED-DFT) has been developed recently[19], enabling the modeling of strong coupling between electromagnetic cavity modes and electronic excitations[37,38]. While the commonly-used TDDFT approaches with classical description of electric fields may not be able to describe the full range of quantum optical phenomena that QED-DFT aims at, we demonstrate in this paper that the cavity mode created by a localized surface plasmon resonance as well as its strong coupling with excitons can be described already within the standard TDDFT. In the following, we argue that theoretical predictions obtained at the nanometer scale may in fact be useful for understanding and modeling of more extended systems. We use TDDFT to study the optical response of aluminum (Al)

nanoparticles composed of a few hundred atoms coupled with one to eight benzene molecules. Thereby we show that cavity quantum electrodynamics holds down to resonators on the order of a few cubic nanometers. In the case of an $Al_{201}$ particle we observe a single-molecule coupling strength of 200 meV due to a very large vacuum field of $4.5\,\mathrm{V\cdot nm^{-1}}$, while for a large number of molecules, the molecules perturb the system increasing the mode volume and decreasing the per-molecule coupling strength. For larger $Al_{586}$ and $Al_{1289}$ nanoparticles the coupling strengths are smaller and follow well the $\sqrt{N}$ dependence of coherent coupling with the number $N$ of molecules.

## Results

**Modeling strong coupling with TDDFT.** We model light-matter interactions by employing the real-time-propagation (RT) TDDFT approach[39] based on the localized basis sets[40,41] as implemented in the open-source GPAW package[42,43]. This RT-TDDFT code is combined with extensive analysis tools[44] that are utilized for analyzing the electron-hole transition contributions to resonances and visualizing them as transition contribution maps (TCM)[44,45]. See Methods for detailed description. We consider a proof-of-principle plasmonic system based on idealized Al nanoparticles. The free-electron-like electronic structure of Al greatly simplifies the analysis in contrast to noble metals with $d$-electron-screened plasmons. In addition, Al has attracted recent interest as an alternative plasmonic material[46].

Our model systems consist of Al nanoparticles (regular truncated octahedra with 201, 586, and 1289 atoms, approximate diameters of 1.8, 2.7, and 3.6 nm, respectively) and benzene molecules (see Supplementary Fig. 1). The $Al_{201}$ nanoparticle exhibits a plasmon resonance at $\hbar\omega_{Al} = 7.7$ eV (Fig. 1a), the collective nature of which is recognizable in the TCM (Fig. 1b)[44]. The benzene molecule has a doubly-degenerate resonance at $\hbar\omega_B = 7.1$ eV with a transition dipole moment $\mu_1 = 4.35$ D (Fig. 1a and Supplementary Fig. 2).

When the nanoparticle and benzene molecules are placed in proximity (bottom part of Fig. 1a) their resonances couple and form two polaritons. The lower (at 6.9 eV) and upper (at 7.7 eV) polaritons are, respectively, mixed symmetric and antisymmetric plasmon-molecule states as apparent in the TCMs of the coupled system (Fig. 1d, e). A comparison with the TCMs of the uncoupled constituents (Fig. 1b, c) reveals the individual contributions to the polaritons. The low-energy ($\lesssim 2$ eV) transitions near the Fermi level form the plasmon, while the contribution from about $-2$ to $3$ eV originates from the molecular exciton. Crucially, the two polaritonic states in Fig. 1d, e differ with respect to the sign of the molecular contributions. In case of the lower polariton (LP) plasmonic and molecular transitions are in-phase, while for the upper polariton (UP) they are out-of-phase. This, respectively, symmetric and antisymmetric combination is also clearly visible in the induced densities (Fig. 1a). At the LP the dipoles of both particle and molecules are parallel, while they are antiparallel at the UP. Such an arrangement is archetypal for a strongly coupled system. This observation is valid also for the other systems discussed below, indicating the presence of strong or near-strong coupling already in the case of a single molecule placed 3 Å away from the $Al_{201}$ particle. We note here, that the density of states of the molecular elements in the coupled system does not significantly differ from its uncoupled counterpart (Supplementary Fig. 3).

To determine the fidelity of TDDFT for modeling strong coupling, we calculated the photoabsorption spectra of $Al_{201}$, $Al_{586}$, and $Al_{1289}$ coupled to $N$ benzene molecules, whose dipole moment is aligned collinearly with that of the plasmon vacuum field. The molecules are placed at the corners of two opposing {100} facets of

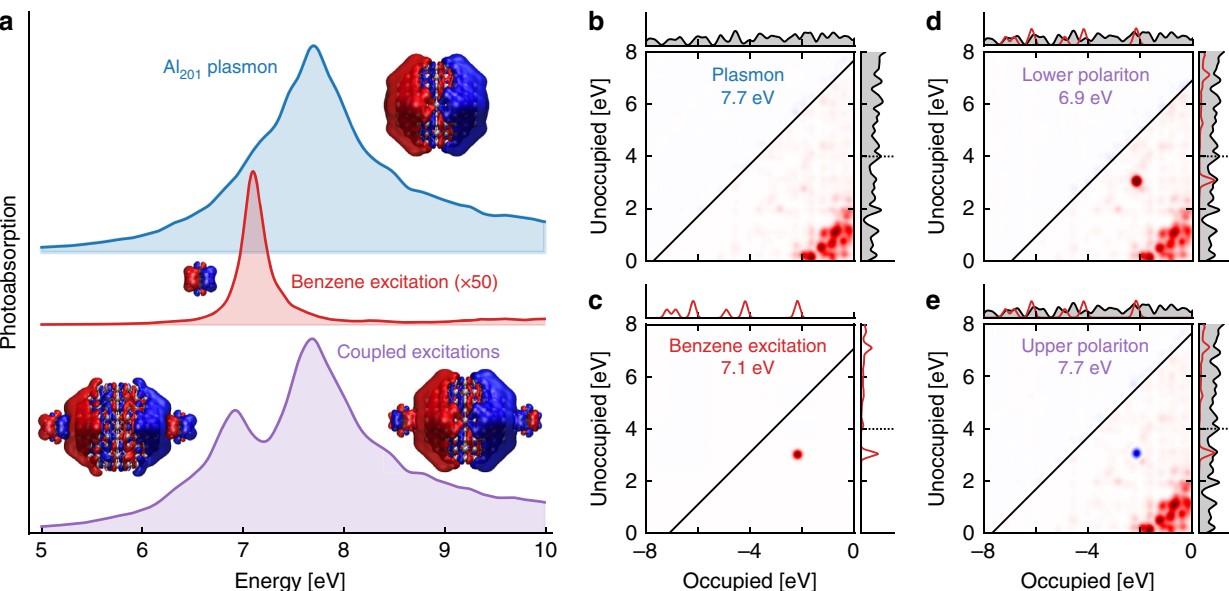

**Fig. 1** Constituent elements of the strongly coupled system. **a** Strong interaction between a collective plasmon excitation of an $Al_{201}$ nanoparticle and a molecular resonance of benzene results in the formation of lower and upper polariton states. Insets show the induced charge densities at resonances. The spectra are to scale but offset for clarity. **b–e** Transition contribution maps (TCMs) of the (**b**) plasmon and (**c**) molecular resonance show, respectively, the collective and discrete nature of these resonances. TCMs for **d** the lower polariton (6.9 eV) and **e** the upper polariton (7.7 eV) show the mixture of plasmonic and molecular transition contributions. The molecular transition (from about −2 to 3 eV) shows a clear symmetric and antisymmetric character with respect to the low-energy (≲2 eV) plasmonic transitions at the lower and upper polaritons, respectively, and the induced densities at the polariton energies visualize the corresponding in-phase and out-of-phase alignments (insets of panel **a**). In the density of states the red line marks the contribution from benzene, multiplied by 5 for visibility, and the vacuum level at around 4 eV is marked by a dashed line

the particles (Supplementary Fig. 1) and structural relaxation is neglected for simplicity. The coupled systems exhibit the LP and UP modes, the splitting of which increases with the number of molecules, see Fig. 2 (TCMs show similar symmetric/antisymmetric mixtures of plasmonic and molecular transitions, see Supplementary Fig. 4). The Rabi splitting of the absorption spectra for $Al_{201}$ and one benzene molecule equals 730 meV, a value comparable to the plasmon width. A non-negligible part of the UP/LP separation originates, however, from the plasmon-exciton detuning and the coupling strength $g$ is 200 meV for one benzene, a value smaller than the geometrical mean of the plasmon and exciton widths (300 meV). The strong coupling condition is, however, fulfilled for $N \geq 3$ for $Al_{201}$ and for larger $N$ for larger particles.

**Coherent coupling and mode volumes.** One signature of collective coherent coupling is the square-root dependence of $g$ on the number $N$ of identical excitons with transition dipole moment $\mu_1$ interacting with an optical cavity in the standard QED expression,

$$g(N) = \sqrt{N}\mu_1 E_{vac}, \qquad (1)$$

where $E_{vac} = \sqrt{\hbar\omega_{Al}/(2\varepsilon_0 V)}$ is the vacuum field and $V$ is the volume of the electromagnetic mode[15]. Using the $Al_{201}$ particle as an example, the expected scaling is shown in Fig. 3 by the black dotted line using $\mu_1$ of a benzene molecule and $E_{vac}^{(201)} = 4.5\,V \cdot nm^{-1}$ with $\hbar\omega_{Al} = 7.7\,eV$ and $V = 3.3\,nm^3$ corresponding to the volume of the $Al_{201}$ particle, which is an adequate estimate for a nanometer-sized particle[47]. To estimate the coupling strengths $g$ from our first-principles data, we fit the absorption spectra with a coupled harmonic oscillator model[48] (see Supplementary Note 1), yielding the $g$ values shown as symbols in Fig. 3. We notice that the coupling strength values obtained using the standard QED expression of Eq. (1), which

is insensitive to spatial variations in the electric field, are larger than the ones derived from coupled oscillator fitting. A similar observation holds for the larger particles $Al_{586}$ and $Al_{1289}$, except, that due to larger volumes, their respective vacuum fields ($E_{vac}^{(586)} = 2.6\,V \cdot nm^{-1}$, $E_{vac}^{(1289)} = 1.8\,V \cdot nm^{-1}$) and coupling strengths are reduced.

**Coupling efficiencies.** Equation (1) is accurate under the condition that all excitonic modes are coherently coupled with the same, maximal rate to the cavity. This is true for structures such as Fabry-Pérot cavities, photonic crystal slabs, or micropillars, in which the anti-node of the mode is accessible to excitonic modes of molecules or semiconductors by precise placement via trapping or doping[3,15]. For isolated plasmonic particles the maximum of its mode is, however, typically inside the particle[47], such as in the present case. Consequently, the molecular coherent dipole moment interacting with the cavity is smaller than its maximum value of 4.35 D, and it is estimated to be in the range from 2 to 4 D, increasing with the size of the nanoparticle. The reduction of the coherent dipole moment is typically expressed in terms of an efficiency factor $\eta$ determined by the ratio of the mode energy density at the position of the exciton to its maximum value[49].

Before extracting efficiency factors $\eta$ from the TDDFT data, we estimate their values on the basis of classical electromagnetic calculations, which are acceptable at a semi-quantitative level for picocavities[50], using a local Drude-permittivity tailored to match the TDDFT absorption spectrum for $Al_{201}$. We find that in the spatial region occupied by the benzene molecule the efficiency $\eta$ ranges from 0.3 to 0.5 (Fig. 4a), due to the rapid decrease of the plasmon-induced electric field. However, such a low value of $\eta$ predicted by calculating the mode profile of a bare cavity might not hold in practice. It is known that the presence of a material with a refractive index greater than the background causes additional field localization, increasing the coupling strength[49].

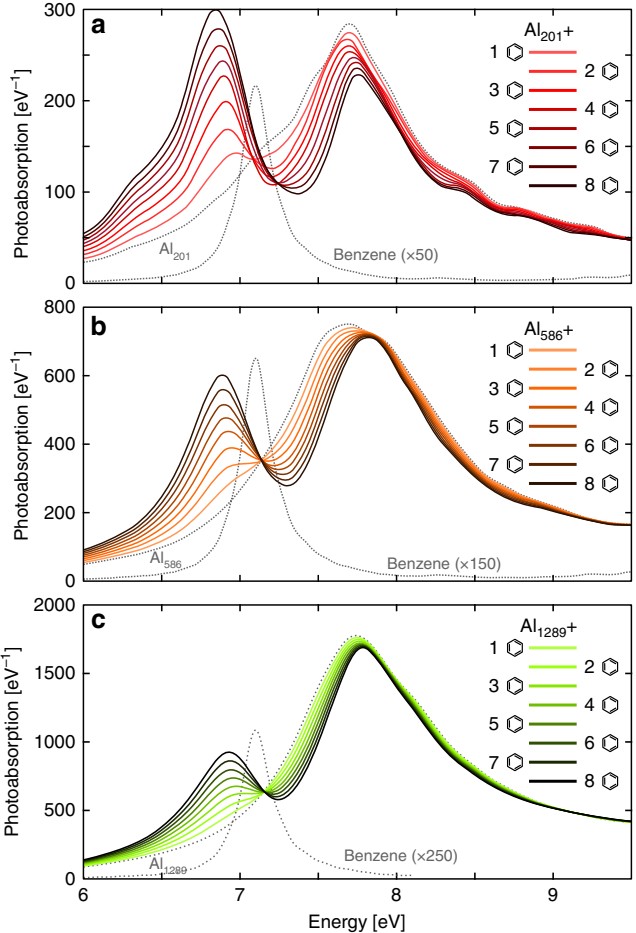

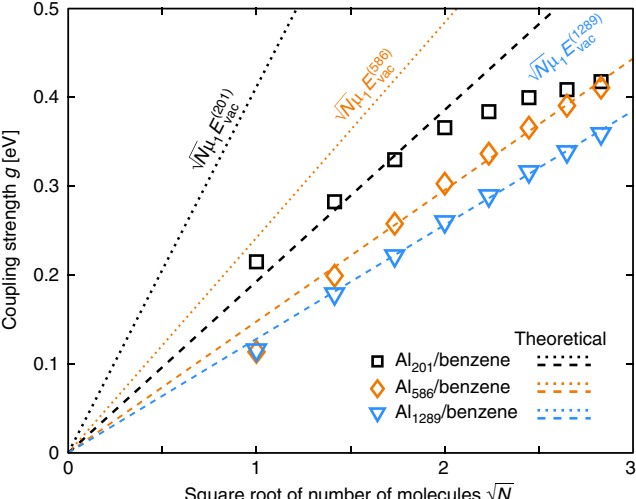

**Fig. 3** Coupling strengths of the nanoparticle-molecule systems. Coupling strengths $g$ for different numbers $N$ of molecules placed 3 Å above the nanoparticle. The symbols denote fitted $g$ values from the data of Fig. 2. The dotted lines mark the ideal theoretical coupling strengths $g = \sqrt{N}\mu_1 E_{vac}$, while the dashed lines mark the ideal dependence multiplied with efficiency factors $\eta$ of $\eta_{201} = 0.47$, $\eta_{586} = 0.61$, and $\eta_{1289} = 0.78$ (see Fig. 4a and its in-text description for details)

**Fig. 2** Strong coupling of plasmons in Al nanoparticles and excitations in benzene molecules. The calculated photoabsorption spectra of nanoparticle-molecule systems composed of (**a**) $Al_{201}$, (**b**) $Al_{586}$, and (**c**) $Al_{1289}$ nanoparticles coupled with $N$ benzene molecules at 3 Å separation (solid lines) show clearly separated lower and upper polaritons, the splitting of which increases with $N$. Spectra of the bare Al particles and benzene molecule are shown for reference (dotted lines)

While a single benzene molecule cannot be meaningfully described in terms of a refractive index, it causes a similar effect. This is supported by the electric near-field enhancements calculated with TDDFT (Fig. 4b–d). In comparison to the field of the plasmon in the bare nanoparticle, in the LP/UP of the coupled system the benzene molecule focuses the electric field, modifying the cavity and, consequently, its vacuum field[49].

To estimate the coupling efficiencies $\eta$ from the TDDFT-derived coupling strengths (Fig. 3), we extract their values as modifications of the slope predicted by Eq. (1). The resulting dependencies $g = \eta\sqrt{N}\mu_1 E_{vac}$ are shown as dashed lines in Fig. 3. For $Al_{201}$, the corresponding value $\eta_{201} = 0.47$ gives a good agreement for $N \leq 3$, but for $N > 3$ the coupling strengths fall below the slope set by $\eta_{201}$. Such a deviation for large $N$ is not observed for the two larger particles, which, respectively, have efficiency factors of $\eta_{586} = 0.61$ and $\eta_{1289} = 0.78$. These larger efficiencies imply that the plasmonic modes of the larger particles are used more efficiently than in the case of $Al_{201}$. Minor deviations seen for the two larger nanoparticles for $N < 3$ are probably caused by inaccurate fitting of the absorption spectra as the LP and UP are not clearly separated.

To test the effect of a diminishing overlap between the vacuum field of the cavity and the molecular exciton, we calculated the dependence of coupling strength on the nanoparticle-molecule separation (Fig. 5). The Rabi splitting of LP and UP decreases with increasing separation and the reduction of the coupling efficiency[49] leads to a decrease in the coupling strength (Fig. 5b).

## Discussion

The coupling efficiencies estimated from classical electromagnetic and TDDFT calculations are consistent. The addition of molecules leads to a systematic red shift of the plasmon (Supplementary Fig. 5), which is largest for $Al_{201}$ and scales approximately inversely with particle volume in agreement with earlier predictions[51]. The red shift happens simultaneously with strong coupling and complicates the analysis. While the calculations yield a $\sqrt{N}$ dependence of the coupling strength for the larger particles ($Al_{586}$, $Al_{1289}$), for the smallest particle ($Al_{201}$) the efficiency varies with the number of benzene molecules for $N \gtrsim 3$. In the $Al_{201}$ case, the volume occupied by the molecules is non-negligible relative to the particle volume and every additional molecule changes the properties of the nanoscale cavity and subsequently modifies the coupling.

From the perspective of classical electromagnetism, the molecules can be represented as a polarizable background with excitons modelled by Lorentzians[52]. This background can be treated as part of the cavity and consequently modifies its local optical density of states (LDoS)[49,53]. This modification manifests itself as focusing of the electric field around the molecules, clearly visible for $Al_{201}$ coupled with one benzene molecule (Fig. 4c, d), in comparison to the resonance of $Al_{201}$ alone (Fig. 4b). These LDoS modifications increase the mode volume, decreasing (or at least not enhancing) the coupling strength per molecule. Such changes of the cavity induced by adding molecules are not visible for larger particles, as for $Al_{586}$ and $Al_{1289}$ the coupling strengths follow the $\sqrt{N}$ dependence.

In conclusion, we demonstrated the suitability of first-principles TDDFT for studying strong coupling between plasmons and molecular excitons. This approach allows us to capture

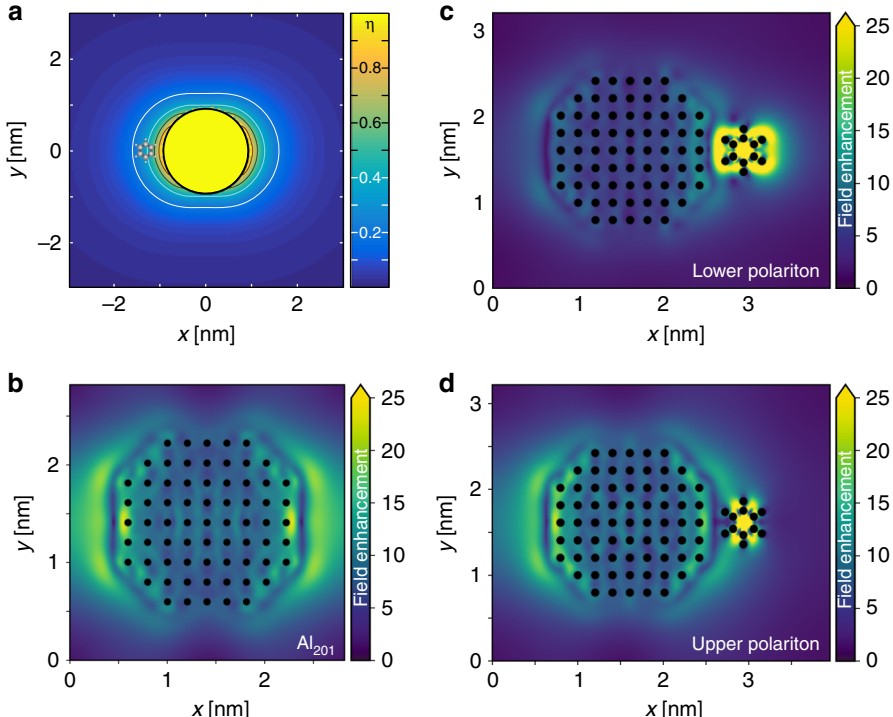

**Fig. 4** Coupling efficiency and electric near-field enhancement in Al$_{201}$. **a** Spatial map of the coupling efficiency (normalized mode energy density, calculated using classical electromagnetism) shows that maximum coupling occurs if the benzene molecule is inside the Al$_{201}$ particle and decreases rapidly away from its surface. At the position of benzene the coupling efficiency is about 30–50% of the maximum value. The contours are spaced every 0.2. **b–d** Field enhancement from time-dependent density-functional theory calculations at (**b**) the Al$_{201}$ resonance and at the Al$_{201}$-benzene (**c**) lower and (**d**) upper polaritons (with benzene located at the center of the facet). Benzene focuses the electric field, perturbing the nanoplasmonic cavity and increasing the coupling efficiency beyond the expectation based on the field at the bare resonance[52]

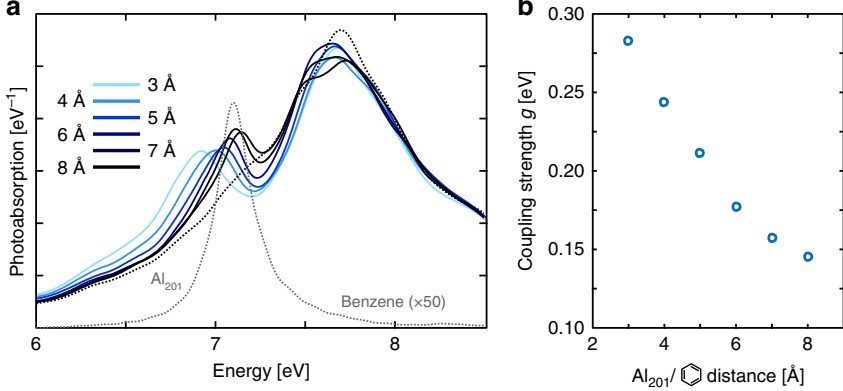

**Fig. 5** Dependence of coupling on nanoparticle-molecule separation. **a** Photoabsorption spectra for two benzene molecules coupled to the Al$_{201}$ particle, showing weaker splitting with increasing separation from 3–8 Å. **b** Coupling strength for two benzene molecules as a function of distance from the Al$_{201}$ particle, showing the expected decrease due to the decreasing overlap integral between exciton and mode

relevant interactions at the atomic level. This is important for studying coupling of molecules to picocavities slightly larger than the molecules themselves[50,54] as well as ultrastrong coupling that leads to modification of the molecular ground state[18,55]. Furthermore, we have shown the degree to which the simple cQED description holds for small systems. For small single-particle cavities, such as the Al$_{201}$ particle considered here, the presence of molecules modifies the cavity and, consequently, its mode volume. The mode volume, in turn, affects the coupling strength, whose value does not increase as quickly as predicted by QED. This slower increase of $g$ is noticeable for the Al$_{201}$ particle, but for larger sizes (Al$_{586}$, Al$_{1289}$) deviations from Eq. (1) become

negligible. Despite the deviations between a simple cQED formula and TDDFT calculations, the order of magnitude agreement between them is quite remarkable. In particular, it is rather surprising that such a simple formula at all holds at such small scales. Based on the obtained coupling strengths, calculated mode volumes and field enhancements, reaching single/few-molecule ultrastrong coupling for single-particle cavities may prove challenging.

## Methods
**DFT and TDDFT calculations**. The DFT and TDDFT calculations were carried out using the Perdew-Burke-Ernzerhof (PBE)[56] exchange-correlation functional in the

adiabatic limit. The spectra were calculated using the $\delta$-kick technique[39] in the linear-response regime and employing the dipole approximation for light-matter interaction. The default projector augmented-wave[57] data sets and double-$\zeta$ polarized (dzp) basis sets provided in GPAW were used. The dzp basis set of Al includes diffuse 3p functions, which are important for describing plasmon resonances[58]. To minimize spurious effects due to the basis-set superposition error, the so-called ghost-atom approach was used separately for each nanoparticle size to keep the total system basis set as intact as possible despite the changing number of surrounding molecules. In general, while the used basis sets might not be adequate for yielding numerical values at the complete-basis-set limit, they are expected to be sufficient for the purposes of the present work. A grid spacing parameter of 0.3 Å was chosen to represent densities and potentials, and the molecules/particles were surrounded by a vacuum region of at least 6 Å. The Hartree potential was evaluated on a larger grid with at least 120 Å vacuum around the system and a coarser grid spacing of 1.2 Å, and subsequently refined to the original grid. For the time propagation, we used a time step of $\Delta t = 15$ as and total propagation time of at least $T = 30$ fs. The spectra were broadened using Lorentzian spectral broadening with $\eta = 0.1$ eV corresponding to a full width at half-maximum of 0.2 eV. Codes for reproducing the calculations are available (see Code availability).

**Transition contribution map (TCM).** A TCM is used for visualizing the Kohn-Sham (KS) electron-hole transition contributions to photoabsorption. Briefly, the photoabsorption cross-section $S(\omega)$ is expressed in the basis of occupied ($i$) and unoccupied ($a$) KS states as $S(\omega) = \sum_{ia} S_{ia}(\omega)$[44]. In TCM, the elements of the matrix $S_{ia}(\omega)$ at a chosen resonance energy are plotted on a Gaussian-broadened two-dimensional plane spanned by the energy axes for occupied and unoccupied KS states. The color indicates the sign of the transition contribution: positive (red) or negative (blue), and the color intensity indicates the magnitude of the contribution. The maps are augmented with the corresponding densities of states, and a diagonal line is drawn to indicate the KS eigenvalue difference corresponding to $\omega$. See ref. [44] for a detailed description of TCM construction.

**Software used.** DFT calculations were carried out using the GPAW package[42,43] with localized basis sets (LCAO mode)[41]. TDDFT calculations were conducted using the LCAO-RT-TDDFT implementation in GPAW[40] and analyzed using the TCM tools therein[44]. The ASE library[59] was used for constructing the atomic structures. The NumPy[60] and Matplotlib[61] Python packages, the VMD software[62,63], and Inkscape were used for processing data and generating figures.

## Data availability

The atomic structures and the photoabsorption spectra are available at https://doi.org/10.5281/zenodo.3242317. The coupling strengths and other fitting parameters are listed in Supplementary Tables 1–4. Other data that support the findings of this study are available from the authors upon reasonable request.

## Code availability

Codes reproducing the data at https://doi.org/10.5281/zenodo.3242317 are available at the same location.

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

## Acknowledgements

We acknowledge financial support from the Swedish Research Council (VR), the Knut and Alice Wallenberg Foundation, the Polish National Science Center via project 2017/25/B/ST3/00744, and the European Union's Horizon 2020 research and innovation programme under the Marie Skłodowska-Curie grant agreement No 838996. T.J.A. acknowledges support from the Project HPC-EUROPA3 (INFRAIA-2016-1-730897), with the support of the EC Research Innovation Action under the H2020 Programme. We acknowledge the computational resources provided by the Swedish National Infra-structure for Computing at PDC, Stockholm, and by the ICM-UW (Grant #G55-6).

## Author contributions

T.P.R. and T.J.A. performed the calculations and data analysis. T.S. and T.J.A. conceived the general approach. T.J.A. coordinated the project. T.P.R., T.S., P.E., and T.J.A. planned the research and wrote the paper.

## Additional information

**Competing interests:** The authors declare no competing interests.

