## [Peer Review File · Nature Communications]

Reviewers' comments:

Reviewer #1 (Remarks to the Author):

The manuscript is devoted to the first principle simulations of the interactions of a molecule and plasmonic system by taking benzene and Al particle as an example. Via their simulations, the authors found that the effective energetics of the system can be understood in terms of upper and lower polaritons.

The topic of molecule-plasmon interactions is studied, e.g., in the works by Prof. Baumberg and others, and the relevant interaction energy scales are already known. Polariton physics is also well developed (see, for example, works of Profs. Yamamoto, Snoke, and others). Also, the collective energy levels could in principle be considered in terms of upper and lower polaritons, however, since the plasmonic system is strongly damped these states have short lifetime and there are doubts if this classification is useful at all.

To conclude, my suggestion is not publish this manuscript in Nature Communications. I recommend to submit this manuscript to a more specialized journal. Thank you.

Reviewer #2 (Remarks to the Author):

This manuscript definitely presents very interesting and exciting results on “first principles” hybrid plasmonic calculations, in particular calculating coupling strengths and strong coupling that is often discussed with more phenomenological approaches such as coupled oscillator models and cavity quantum electrodynamics models. It also presents interesting results in and of themselves for the aluminum nanoparticle/benzene systems. I would say this manuscript should be published in Nature Communications once the authors address the following minor points.

1. The “ \sqrt{N} ” dependence is noted to be a signature of strong coupling – see eq (1) and discussion. However that type of dependence arises from Tavis-Cummings or Dicke models – see Garraway, ref. [17], eq (2.9). All that is needed is to have N systems interacting independently of one another but each one coupling directly with the resonator. More simply, in such a limit one might expect a total rate = N * (the base 1-system rate), and since rates are squares of transition matrix elements, the corresponding effective coupling element is \sqrt{N} * (the base 1-system coupling).

2. It would be helpful to have the sizes or effective radii of the various Aluminum nanoparticles noted.

3. It would also be interesting to see how well simple Mie theory, with a standard dielectric constant (such as that of Rakic et al.) used for Al compares with at least the isolated nanoparticle resonances. I made some rough estimates of the sizes and appear to be finding a resonance of 9 eV for the 201 atom case, somewhat higher than the 7.7 eV TDDFT result but – my size estimate could have been quite off or the classical limit is poor in this case.

Reviewer #3 (Remarks to the Author):

To the best of my knowledge this paper presents the first calculation using first-principles electronic

structure calculations of strong coupling between a molecular exciton and the plasmon resonance of a metallic particle. The authors use state of the art methods based on the so-called time dependent density functional theory (TDDFT) to compute the plasmon-molecule coupling of a system composed by aluminum nanoparticles (containing 201, 586 and 1289 atoms) with benzene molecules (from 1 to 8 molecules) in their neighborhood. The analysis of the results permits to obtain the coupling strength g as a function of the size of the nanoparticle, the number of molecules and the distance of the molecules to the particle surface. The results confirm the expectations from analysis using simple models of macroscopic cavity quantum electrodynamics, in particular the square root dependence (\sqrt{N}) of g with the number of molecular emitters (N).

Given the large attention that the strong coupling regime is attracting in recent times in the field of nanooptics and nanoplasmonics, the novelty of the results (to best of my knowledge this is the first fully microscopic, atomistic calculations without empirical input of molecular exciton-plasmon coupling in a plasmonic cavity) and the high quality of the presentation I recommend the publication of the paper in Nature Communications after they have reviewed their paper according to my comments/criticisms below.

I have a few comments for the authors in order to improve the paper:

1) I am somewhat concerned about some of the comments made by the authors claiming that their approach can tackle effects "including ground state, chemical and thermodynamic modifications of the molecules in the strong regime,...". This claim appears mostly in the last sentence of the abstract. I think this is misleading. One must be very cautious here. The description of the electromagnetic field in the present calculations is completely classical and the effect of quantum fluctuations in the ground state of the system (both of the field and the electronic density of the objects) are NOT included. Under such circumstances, renormalization of the ground state or modifications of the chemical properties of the system inside a cavity will not be captured by the level of theory used in the present paper. The authors should be very clear and honest about this. In particular, they should eliminate or modify accordingly the statement in the abstract and any other statement in the paper that could potentially create such confusion about the capability of the used approach (hopefully in the future some these additional effects will be include through QED-DFT).

-The definition of the distance between the molecule and the nanoparticle should be given with more detail. This is a very important parameter for the intensity of the coupling, but also for the comparison to experiment. How is the distance defined? between the closest atom (including H) of the molecule and the surface atoms of the particle?

Other comments:

-Caption of Fig. 1: "... the induced energy density at these energies...". I guess it is meant: "... the induce density at these energies...".

-The first row in Table S4 in the supporting information (for Al201, 2 benzene molecules and varying distance) correspond to a molecule-particle distance of 3 Ang. Thus, I assumed that the result should be the same as the second row in Table S1. However, there are some small differences in the values of the resonance positions and coupling strength. Could the authors explain why is so? Clarifying that in the SI could be interesting for the reader trying to reproduce or understand in depth the presented results.

With best regards,
Daniel Sánchez-Portal

Reviewer #1 (Remarks to the Author)

The manuscript is devoted to the first principle simulations of the interactions of a molecule and plasmonic system by taking benzene and Al particle as an example. Via their simulations, the authors found that the effective energetics of the system can be understood in terms of upper and lower polaritons.

The topic of molecule-plasmon interactions is studied, e.g., in the works by Prof. Baumberg and others, and the relevant interaction energy scales are already known. Polariton physics is also well developed (see, for example, works of Profs. Yamamoto, Snoke, and others). Also, the collective energy levels could in principle be considered in terms of upper and lower polaritons, however, since the plasmonic system is strongly damped these states have short lifetime and there are doubts if this classification is useful at all.

To conclude, my suggestion is not publish this manuscript in Nature Communications. I recommend to submit this manuscript to a more specialized journal. Thank you.

Authors' reply:

We thank the Reviewer for taking the time to read through our manuscript. We disagree, however, with the conclusion. We acknowledge that the earlier works by Profs. Baumberg, Yamamoto, Snoke and others are seminal works. Yet the Reviewer appears to disregard the crucial fact that our work demonstrates a *quantitative and predictive* means to predict the relevant energy scales without relying on any assumptions regarding interaction parameters or approximate calculations (such as classical electromagnetic simulations, model Hamiltonians or phenomenological approaches). In this respect, our work achieves a considerable advancement of the field.

Reviewer #2 (Remarks to the Author)

This manuscript definitely presents very interesting and exciting results on first principles hybrid plasmonic calculations, in particular calculating coupling strengths and strong coupling that is often discussed with more phenomenological approaches such as coupled oscillator models and cavity quantum electrodynamics models. It also presents interesting results in and of themselves for the aluminum nanoparticle/benzene systems. I would say this manuscript should be published in Nature Communications once the authors address the following minor points.

Authors' reply:

We thank the Reviewer for the positive and constructive comments. We hope our answers given below fully address all issues.

1. The \sqrt{N} dependence is noted to be a signature of strong coupling see eq (1) and discussion. However that type of dependence arises from Tavis-Cummings or Dicke models see Garraway, ref. [17], eq (2.9). All that is needed is to have N systems interacting independently of one another but each one coupling directly with the resonator. More simply, in such a limit one might expect a total rate = N^ (the base 1-system rate), and since rates are squares of transition matrix elements, the corresponding effective coupling element is \sqrt{N}^* (the base 1-system coupling).*

Authors' reply:

We completely agree with the Reviewer that the \sqrt{N} -dependence appears as a result of Tavis-Cummings or Dicke Hamiltonians, and so it is in fact not necessary to reach the strong coupling regime. However, what we mean here is that the observed Rabi splitting in our TDDFT calculations scales as \sqrt{N} , which is a consequence of collective coherent strong coupling process (which, of course, still can be treated by a Tavis-Cummings Hamiltonian). We have added the phrase "collective coherent" to the sentence preceding eq. (1).

2. It would be helpful to have the sizes or effective radii of the various Aluminum nanoparticles noted.

Authors' reply:

The approximate diameters of the Al nanoparticles are as follows (estimated as a distance between the furthestmost atoms):

Al₂₀₁: 1.8 nm
Al₅₈₆: 2.7 nm
Al₁₂₈₉: 3.6 nm

These values have been added to the main text of our manuscript.

3. It would also be interesting to see how well simple Mie theory, with a standard dielectric constant (such as that of Rakic et al.) used for Al compares with at least the isolated nanoparticle resonances. I made some rough estimates of the sizes and appear to be finding a resonance of 9 eV for the 201 atom case, somewhat higher than the 7.7

eV TDDFT result but my size estimate could have been quite off or the classical limit is poor in this case.

Authors' reply:

For isolated nanospheres in the classical quasistatic limit the dipolar surface plasmon resonance is at $\omega_p/\sqrt{3}$, where ω_p is the plasma frequency of the metal in question. Most literature values for the plasma frequency of Al are in the range 14 to 15 eV, translating to a plasmon resonance at approximately 8.1 to 8.7 eV in the quasistatic limit. The lower end of this range is very close to our calculations, while the upper end is closer to the value quoted by the Reviewer. This apparent discrepancy results from the fact that a local permittivity is inadequate to model such small nanoparticles. Nonlocality, spill-out, and quantum size effects should be considered as well and for Al these will bring the resonance energy down. Examples of these effects on the surface plasmon resonance can be found in New J. Phys. 15 083044 (2013) as well as references therein.

Reviewer #3 (Remarks to the Author)

To the best of my knowledge this paper presents the first calculation using first-principles electronic structure calculations of strong coupling between a molecular exciton and the plasmon resonance of a metallic particle. The authors use state of the art methods based on the so-called time dependent density functional theory (TDDFT) to compute the plasmon-molecule coupling of a system composed by aluminum nanoparticles (containing 201, 586 and 1289 atoms) with benzene molecules (from 1 to 8 molecules) in their neighborhood. The analysis of the results permits to obtain the coupling strength g as a function of the size of the nanoparticle, the number of molecules and the distance of the molecules to the particle surface. The results confirm the expectations from analysis using simple models of macroscopic cavity quantum electrodynamics, in particular the square root dependence (\sqrt{N}) of g with the number of molecular emitters (N).

Given the large attention that the strong coupling regime is attracting in recent times in the field of nanooptics and nanoplasmonics, the novelty of the results (to best of my knowledge this is the first fully microscopic, atomistic calculations without empirical input of molecular exciton-plasmon coupling in a plasmonic cavity) and the high quality of the presentation I recommend the publication of the paper in Nature Communications after they have reviewed their paper according to my comments/criticisms below.

Authors' reply:

We thank the Reviewer for the positive assessment of our work and the constructive comments.

I have a few comments for the authors in order to improve the paper: 1) I am somewhat concerned about some of the comments made by the authors claiming that their approach can tackle effects "including ground state, chemical and thermodynamic modifications of the molecules in the strong regime,...". This claim appears mostly in the last sentence of the abstract. I think this is misleading. One must be very cautious here. The description of the electromagnetic field in the present calculations is completely classical and the effect of quantum fluctuations in the ground state of the system (both of the field and the electronic density of the objects) are NOT included. Under such circumstances, renormalization of the ground state or modifications of the chemical properties of the system inside a cavity will not be captured by the level of theory used in the present paper. The authors should be very clear and honest about this. In particular, they should eliminate or modify accordingly the statement in the abstract and any other statement in the paper that could potentially create such confusion about the capability of the used approach (hopefully in the future some these additional effects will be include through QED-DFT).

Authors' reply:

This is a very important point that is made by the Reviewer with which we agree. While the referred sentence was intended to be an "outlook" for the future and the QED-DFT approach was explained in the main text, we thank the Reviewer for noticing that the sentence in the abstract could be seriously misinterpreted. Correspondingly, we have improved the abstract in that regard.

-The definition of the distance between the molecule and the nanoparticle should be given with more detail. This is a very important parameter for the intensity of the coupling, but also for the comparison to experiment. How is the distance defined? between the closest atom (including H) of the molecule and the surface atoms of the particle?

Authors' reply:

The molecules were placed above the respective facets of the Al nanoparticles parallel to the facet normals. The distance is defined as the separation between the location of the Al facet over which the molecule is placed and the location of the closest hydrogen atom(s). An exemplary sketch of this definition was added to Figure S4.

Other comments: -Caption of Fig. 1: "... the induced energy density at these energies...". I guess it is meant: "... the induced density at these energies...".

Authors' reply:

Indeed. We thank the Reviewer for noticing this mistake.

-The first row in Table S4 in the supporting information (for Al₂₀₁, 2 benzene molecules and varying distance) correspond to a molecule-particle distance of 3 Å. Thus, I assumed that the result should be the same as the second row in Table S1. However, there are some small differences in the values of the resonance positions and coupling strength. Could the authors explain why is so? Clarifying that in the SI could be interesting for the reader trying to reproduce or understand in depth the presented results.

Authors' reply:

The physical systems in both cases are, indeed, the same. The small differences between the two results presented in the tables are due to the fitting procedure. We aimed at minimizing the number of free parameters to prevent overfitting. For Al₂₀₁ with variable number of molecules N this meant fixing the plasmon and exciton widths. This yielded reasonable fitting results. However, for the distance dependence case the exciton width had to be treated as a fitting parameter and we decided to use the same constraints in this batch. This different fitting treatment is the reason for the small differences seen in Tables S1 (for $N = 2$) and S4 (for 3 Å).

REVIEWERS' COMMENTS:

Reviewer #2 (Remarks to the Author):

The authors have adequately addressed the reviewers' concerns and the paper is now acceptable for publication in Nature Comm.

Reviewer #3 (Remarks to the Author):

I am satisfied with the responses of the authors to my comments and those of the other referees. In my opinion the paper is now ready for publication in Nature Communications.

Reviewer #2 (Remarks to the Author)

The authors have adequately addressed the reviewers' concerns and the paper is now acceptable for publication in Nature Comm.

Authors' reply:

We thank the Reviewer for taking the time to comment our manuscript and helping us improve it.

Reviewer #3 (Remarks to the Author)

I am satisfied with the responses of the authors to my comments and those of the other referees. In my opinion the paper is now ready for publication in Nature Communications.

Authors' reply:

We thank the Reviewer for the positive opinion of our changes to the manuscript and recommendation to publish our work.